# A simple RANS inflow model of the neutral and stable atmospheric boundary layer applied to wind turbine wake simulations

Maarten Paul van der Laan[1], Mark Kelly[1], Mads Baungaard[1], Antariksh Dicholkar[1], and Emily Louise Hodgson[1]

[1]Technical University of Denmark, DTU Wind and Energy Systems, Risø Campus, Frederiksborgvej 399, 4000 Roskilde, Denmark

**Correspondence:** Maarten Paul van der Laan (plaa@dtu.dk)

**Abstract.** Wind turbines are increasing in size and operate more frequently above the atmospheric surface layer, which requires improved inflow models for numerical simulations of turbine interaction. In this work, a steady-state Reynolds-averaged Navier-Stokes (RANS) model of the neutral and stable atmospheric boundary layer (ABL) is introduced. The model incorporates buoyancy in the turbulence closure equations using a prescribed Brunt-Väisälä frequency, does not require a global turbulence length scale limiter, and is only dependent on two non-dimensional numbers. Assuming a constant temperature gradient over the entire ABL, although a strong assumption, leads to a simple and well-behaved inflow model. RANS wake simulations are performed for shallow and tall ABLs, and the results show good agreement with large-eddy simulations in terms of velocity deficit from a single wind turbine. However, the proposed RANS model underpredicts the magnitude of the velocity deficit of a wind turbine row for the shallow ABL case. In addition, RANS ABL models can suffer from numerical problems when they are applied as a shallow ABL inflow model to large wind farms, due to the low eddy viscosity layer above the ABL. The proposed RANS model inherits this issue and further research is required to solve it.

## 1 Introduction

Wind turbine and farm interaction can lead to energy losses and increased turbine loads, mainly due to wakes from upstream turbines and farms, but also because of blockage effects (Porté-Agel et al., 2020). The magnitude of these effects is strongly influenced by the atmospheric conditions such as ambient turbulence intensity (Nilsson et al., 2015), buoyancy and boundary layer depth (Hansen et al., 2012). Traditionally, models for simulating wake losses assume simple atmospheric conditions that only represent the first 10% of the atmospheric boundary layer (ABL), known as the atmospheric surface layer (ASL). Examples are wind speed profiles based on a power law or Monin-Obukhov Similarity Theory (Monin and Obukhov, 1954). However, wind turbines are increasing in size and operate more frequently above the ASL, especially for shallow ABLs. Hence, there is a need for improved inflow models that can capture the effect of the ABL on the wind farm flow.

Wind farm flow models based on Computational Fluid Dynamics (CFD) can be employed to simulate wake losses (Porté-Agel et al., 2020). High-fidelity turbulence-resolving and transient CFD methods as large-eddy simulation (LES) is a popular method in academia because it can simulate the complex interaction between the ABL and a wind farm; however, it is too

expensive to simulate all wind direction and wind speed flow cases that are necessary to calculate wake losses in terms of annual energy production (AEP). For the latter, the industry employs engineering wake models because of their computational speed. However, such models require calibration and are often not general enough to perform well for a wide range of atmospheric conditions and wind farm layouts due to the need for assuming a single wake shape and wake superposition method. Reynolds-averaged Navier-Stokes (RANS) is a medium-fidelity steady-state CFD method that is several orders of magnitude faster than LES and does not require the engineering wake model assumptions. An idealized RANS setup of a large wind farm ($16 \times 16$ turbines with $8D$ inter spacing) can simulate AEP wake losses in roughly a day using 624 CPUs (van der Laan et al., 2022). However, RANS requires a turbulence model, which is not trivial, but reasonable results in terms of velocity and power deficits can be achieved (Politis et al., 2012; van der Laan et al., 2015c, b; Baungaard et al., 2022b). In addition, atmospheric inflow modeling in RANS is challenging because the inflow needs to be a solution of the RANS model and numerical convergence is not guaranteed when ABL models beyond the neutral ASL are employed as an inflow model to wind farm simulations (van der Laan et al., 2023b).

Transient ABL models can be employed for inflow to complex terrain and wind farm simulations via unsteady RANS (URANS) (Koblitz et al., 2015; Castro et al., 2015). Such model setups typically include buoyancy terms in the momentum and turbulence transport equations that are linked to an *active* temperature equation. However, URANS has significant disadvantages, starting with the need to solve in time, and the inflow developing downstream; the former requires computational time an order of magnitude larger than RANS, and the latter results in non-trivial complications in model-setup to obtain the desired inflow at a given wind farm location. Instead of using URANS, some authors use a RANS setup by including an inflow that is not a steady-state solution of the employed RANS model, by including e.g. an active temperature equation (Bleeg et al., 2015; Quick et al., 2024). In that case the same issue of non-stationary and horizontally inhomogeneous inflow is encountered, which introduces the distance between upwind domain edge and wind farm as a parameter upon which the results depend.

Steady-state ABL inflow models generally rely on the global length scale limiter of Apsley and Castro (1997), where a maximum turbulence length scale is chosen that indirectly determines the ABL height. Neutral or stable atmospheric conditions can be represented by setting relatively large or small values of the maximum turbulence length scale, to obtain tall or shallow ABLs, respectively. However, unstable conditions, i.e., convective ABLs (CBLs), cannot be modeled without additional model components; it is not trivial to obtain realistic results in the surface layer without getting nonphysical ABL heights (van der Laan et al., 2020). One can argue that CBLs are inherently unsteady, which challenge steady-state model prescription; therefore the present article focuses on neutral and stable atmospheric conditions. When an inflow model based on the global turbulence length scale limiter of Apsley and Castro (1997) is applied to a 3D RANS simulation (Koblitz et al., 2015; Arroyo et al., 2014; van der Laan et al., 2015a; Avila et al., 2017; Ivanell et al., 2018; Freitas et al., 2024), as for a wind farm or complex terrain, then all turbulence length scales will also be limited, which can result in non physical solutions. In previous work (van der Laan et al., 2023b), an alternative ABL inflow model was proposed, where the global turbulence length scale limiter of Apsley and Castro (1997) was replaced by a turbulent buoyant-destruction term, using a prescribed potential temperature profile to represent conventionally neutral ABLs. This model works well for tall ABLs but can have problems for shallow ABLs (as shown later in Appendix A). In the present work, a new ABL inflow model is proposed that does not require a global length

scale limiter, and which is further suited to model stable *and shallow* ABLs. The model employs a turbulent buoyancy source
that depends on a prescribed Brunt-Väisälä frequency by assuming a constant temperature gradient over the entire ABL. While
this is a strong assumption, the resulting model is simple and well-behaved. In addition, the proposed model can simulate
the effect of neutral and stable atmospheric conditions on a wind turbine wake in RANS. The two existing and the proposed
RANS inflow models are discussed in detail in Sect. 2. The three RANS inflow models are applied to single wake simulations
following a methodology described in Sect. 3, and the results are compared with results of LES in Sect. 4, for both a shallow
and a tall ABL. The shallow ABL case is also applied to a wind turbine row.

## 2 RANS inflow models of the ABL

RANS inflow models of the ABL are based on a numerical solution of the 1D momentum equations, for streamwise and lateral
velocity components, $U$ and $V$, respectively, including a prescribed pressure gradient in the form of a geostrophic wind speed,
$G = \sqrt{U_G^2 + V_G^2}$, and Coriolis forces. Here, $U_G$ and $V_G$ are the streamwise and lateral component of geostrophic wind vector.
The momentum equations only depend on a single Cartesian coordinate, namely, the vertical coordinate, $z$:

$$f_c(V - V_G) + \frac{d}{dz}\left(\nu_T \frac{dU}{dz}\right) = 0, \qquad -f_c(U - U_G) + \frac{d}{dz}\left(\nu_T \frac{dV}{dz}\right) = 0, \tag{1}$$

with $f_c$ as the Coriolis parameter. In addition, we have employed the Boussinesq hypothesis with $\nu_T$ as the eddy viscosity for
which a turbulence model is required. In the present work, we use the $k$-$\varepsilon$-$f_P$ eddy viscosity model (van der Laan et al., 2015c)
that employs a transport equation for both the turbulent kinetic energy, $k$, and its dissipation, $\varepsilon$:

$$\nu_T = C_\mu f_P \frac{k^2}{\varepsilon}, \tag{2}$$

$$\frac{d}{dz}\left(\frac{\nu_T}{\sigma_k}\frac{dk}{dz}\right) + \mathcal{P} - \varepsilon + \mathcal{B} + S_{k,\mathrm{amb}} = 0, \qquad \frac{d}{dz}\left(\frac{\nu_T}{\sigma_\varepsilon}\frac{d\varepsilon}{dz}\right) + \left(C_{\varepsilon,1}^* \mathcal{P} - C_{\varepsilon,2}\varepsilon + C_{\varepsilon,3}\mathcal{B}\right)\frac{\varepsilon}{k} + S_{\varepsilon,\mathrm{amb}} = 0,$$

with $f_P$ as a scalar function that acts as a local turbulence length scale limiter in regions with high velocity gradients to assure
realizable Reynolds stresses, which is mainly applicable to a wind turbine (near) wake. However, $f_P$ is not of importance to an
inflow model but applied to be consistent with a 3D RANS simulation of a wind turbine wake. Furthermore, $\mathcal{P}$ and $\mathcal{B}$ are the
turbulent production due to shear and buoyancy, respectively:

$$\mathcal{P} = \nu_T\left[\left(\frac{dU}{dz}\right)^2 + \left(\frac{dV}{dz}\right)^2\right], \qquad \mathcal{B} = \frac{g}{\theta_0}\overline{\theta'w'} = -\frac{\nu_T}{\sigma_\theta}\frac{g}{\theta_0}\frac{d\Theta}{dz}, \tag{3}$$

with $g = 9.81 \text{ ms}^{-2}$ as the magnitude of the gravitational acceleration vector, $\Theta$ as the mean potential temperature with $\theta_0$ as the
hydrostatic background temperature (here we use the value at the wall boundary), and a simple flux-gradient relationship for
the heat flux, $\overline{\theta'w'} = -(\nu_T/\sigma_\theta)d\Theta/dz$, is employed. Note that in order to obtain a steady-state solution of the ABL, one cannot
employ an active temperature equation in combination with a non-linear temperature profile, as such a setup would effectively
become an unsteady RANS method due to a forever growing ABL height. $S_{k,\mathrm{amb}}$ and $S_{\varepsilon,\mathrm{amb}}$ are additional source terms used
to maintain a small ambient value of turbulence for numerical robustness such that $k = k_{\mathrm{amb}}$ and $\varepsilon = \varepsilon_{\mathrm{amb}}$ in absence of any

velocity gradients, applicable to the flow above the ABL (van der Laan et al., 2015a):

$$S_{k,\mathrm{amb}} = \varepsilon_{\mathrm{amb}}, \qquad S_{\varepsilon,\mathrm{amb}} = C_{\varepsilon,2} \frac{\varepsilon_{\mathrm{amb}}^2}{k_{\mathrm{amb}}}, \qquad k_{\mathrm{amb}} = \frac{3}{2} G^2 I_{\mathrm{amb}}^2, \qquad \varepsilon_{\mathrm{amb}} = C_\mu^{3/4} \frac{k_{\mathrm{amb}}^{3/2}}{\ell_{\mathrm{amb}}}, \tag{4}$$

with $\ell_{\mathrm{amb}}$ and $I_{\mathrm{amb}}$, as the ambient turbulence length scale and turbulence intensity (based on $k$) above the ABL, respectively, and $C_{\mathrm{amb}}$ as a model constant (van der Laan et al., 2020). The values of $I_{\mathrm{amb}}$ and $C_{\mathrm{amb}}$ are set small enough to not influence the inflow model solution. Furthermore, the definition of $\ell_{\mathrm{amb}}$ differs with the chosen inflow model and is discussed in Sects. 2.1-2.3. In addition, the following turbulence model constants are used: $(C_\mu, C_{\varepsilon,1}, C_{\varepsilon,2}, \sigma_k, \sigma_\varepsilon, \sigma_\theta) = (0.03, 1.21, 1.92, 1.0, 1.3, 0.74)$, and turbulence model parameters $C_{\varepsilon,1}^*$ and $C_{\varepsilon,3}$ are also discussed in Sects. 2.1-2.3.

## 2.1 RANS-$\ell_{\mathrm{max}}$: Inflow model using the turbulence length scale limiter of Apsley and Castro (1997)

The global turbulence length scale limiter of Apsley and Castro (1997) can be employed to model a neutral and a stable inflow model without the need for turbulent buoyancy source term ($\mathcal{B} = 0$). The limiter represents a variable $C_{\varepsilon,1}^*$ in the transport equations of $\varepsilon$:

$$C_{\varepsilon,1}^* = C_{\varepsilon,1} + (C_{\varepsilon,2} - C_{\varepsilon,1}) \frac{\ell}{\ell_{\mathrm{max}}} \tag{5}$$

where $\ell \equiv C_\mu^{3/4} k^{3/2} / \varepsilon$ is a model-based turbulence length scale. When $\ell$ exceeds $\ell_{\mathrm{max}}$ then the source terms in the $\varepsilon$ equation cancel and this prevents the turbulence length scale from growing larger than the maximum set value, $\ell_{\mathrm{max}}$. The height of the ABL can be set implicitly using $\ell_{\mathrm{max}}$. For $\ell_{\mathrm{max}} \to 0$ and $\ell_{\mathrm{max}} \to \infty$, the analytic ABL solutions of Ekman (1905) (constant $\nu_T$) and Ellison (1956) (linear $\nu_T$ with $z$) are obtained, respectively, which bounds the numerical RANS model, as shown in van der Laan et al. (2020). When the global turbulence length scale limiter of Apsley and Castro (1997) is applied as an inflow model to a 3D RANS simulation, then all turbulence length scales are limited and this can lead to a non-physical recovery of a wake generated by for example a wind turbine, a wind farm or a hill (Koblitz et al., 2015; van der Laan et al., 2015a; Avila et al., 2017). An ad-hoc solution has been proposed in previous work (van der Laan et al., 2015a) by switching off the turbulence length scale limiter in wake region using the $f_P$ function as a wake identifier:

$$C_{\varepsilon,1}^* = f_1 \left[ C_{\varepsilon,1} + (C_{\varepsilon,2} - C_{\varepsilon,1}) \frac{\ell}{\ell_{\mathrm{max}}} \right], \qquad f_1 = \frac{1}{2} \left[ \tanh \left( 50 \left[ f_P - 0.9 \right] \right) + 1 \right] \tag{6}$$

Here, $f_1$ is a blending function that switches between the global ($\ell_{\mathrm{max}}$) and local ($f_P$) turbulence length scale limiters. The impact of this solution on a single wake is further investigated in Sect. 4. The ambient values of $k$ and $\varepsilon$ are set by Eq. (4), where the ambient turbulence length scale is defined as:

$$\ell_{\mathrm{amb}} = C_{\mathrm{amb}} \ell_{\mathrm{max}} \tag{7}$$

with $C_{\mathrm{amb}} = 10^{-6}$ and $I_{\mathrm{amb}} = 10^{-6}$ (van der Laan et al., 2020). We label the inflow model as the RANS-$\ell_{\mathrm{max}}$ model.

## 2.2 RANS-$\Theta$: Prescribed temperature inflow model

The RANS-$\ell_{\mathrm{max}}$ can lead to non-physical wake recovery when it is applied as an inflow model to wind farm, especially for shallow ABLs. To overcome this issue, an alternative RANS inflow model has recently been developed (van der Laan et al.,

2023b), here labeled as the RANS-$\Theta$ model, where the global length scale limiter of Apsley and Castro (1997) has been replaced ($C_{\varepsilon,1}^* = C_{\varepsilon,1}$) by the use of a non-zero turbulence buoyancy from Eq. (3), and an analytic prescribed temperature profile that includes a constant temperature in the surface layer and a constant inversion:

$$\frac{d\Theta}{dz} = \frac{1}{2}\left[1 + \tanh\left(\frac{z/z_i - 1}{z_T/z_i}\right)\right]\left.\frac{d\Theta}{dz}\right|_c, \tag{8}$$

where $z_i$ is the inversion height, $d\Theta/dz|_c$ is the inversion strength, and $z_T$ characterizes the distance over which the temperature gradient changes from 0 to $d\Theta/dz|_c$ (we take $z_T/z_i = 0.2$). The temperature profile can be obtained upon integration and its final form is described in van der Laan et al. (2023b). The temperature profile remains constant when the model is applied as inflow to a 3D RANS simulation, since $\Theta(z)$ from Eq. (8) is prescribed instead of solving a temperature equation. Note that the original RANS-$\Theta$ model was employed with a slightly different implementation of the buoyancy compared to Eq. (3), namely, $\mathcal{B} = -(\nu_T/\sigma_\theta)(g/\Theta)d\Theta/dz$. However, $\Theta \simeq \theta_0$, since $z_i d\Theta/dz|_c \ll \theta_0$ for the values of $z_i$ and $d\Theta/dz|_c$ encountered in the ABL (permitting us to also use the wall temperature for $\theta_0$).

The ambient turbulence length scale above the ABL is defined as

$$\ell_{\mathrm{amb}} = C_{\mathrm{amb}} z_i. \tag{9}$$

In addition, we use $I_{\mathrm{amb}} = 10^{-5}$ and $C_{\mathrm{amb}} = 10^{-7}$. Finally, the $C_{\varepsilon,3}$ constant is defined as

$$C_{\varepsilon,3} = 1 + C_{\varepsilon,1} - C_{\varepsilon,2} \tag{10}$$

following Sogachev et al. (2012) for $\ell_{\max} \to \infty$.

The RANS-$\Theta$ model is suited to model a conventionally neutral ABL (CNBL). However, if one selects an inconsistent combination of $z_i$ and $d\Theta/dz|_c$ then an unphysical inflow profile (with effectively two ABL heights) can result. This problem is further illustrated in Appendix A for a (too) shallow ABL.

## 2.3 RANS-$N$: New inflow model based on a constant Brunt-Väisälä frequency

We propose to write the buoyant destruction of turbulent kinetic energy from Eq. (3) as

$$\mathcal{B} = -\frac{\nu_T}{\sigma_\theta}\frac{g}{\theta_0}\frac{d\Theta}{dz} = -\frac{\nu_T}{\sigma_\theta}N^2; \tag{11}$$

the Brunt-Väisälä frequency is described by

$$N \equiv \sqrt{\frac{g}{\theta_0}\frac{d\Theta}{dz}}. \tag{12}$$

The Brunt-Väisälä frequency is a measure of stable stratification, normally applied to the inversion layer of the ABL or 'free atmosphere' above.

The problems with the RANS-$\ell_{\max}$ and RANS-$\Theta$ models outlined above can be overcome by prescribing a constant gradient of temperature throughout the entire ABL in Eq. (11), giving a constant $N \to N_{\mathrm{ABL}}$ in Eq. (12). The turbulence model constant

$\sigma_\theta$ (turbulent Prandtl number) is set to one for simplicity, as it could be absorbed into $N_{\mathrm{ABL}}$. The RANS-$\Theta$ model can also be written in the form of (11), but with a vertically varying temperature gradient and $N(z)$, where as $z \to z_i$ in the upper ABL $d\Theta/dz \to (d\Theta/dz)|_c$ and $N \to N_c$. The simple form of $\mathcal{B}$ with a constant $N$ also implies that the heat flux profile is same as the eddy viscosity profile times a constant, $\overline{\theta' w'} = -N^2(\theta_0/[g\sigma_\theta])\nu_T$. A constant temperature gradient was also assumed by

150 Chougule et al. (2017) to simulate atmospheric boundary turbulence with a spectral tensor model including effects of buoyancy. Using a constant $N$ or constant temperature gradient for the entire ABL is not always realistic, but this model choice results in a simple RANS ABL inflow model, which we label the RANS-$N$ model, that can yield reasonable results of the ABL; this is further discussed in Sect. 4. Furthermore, the RANS-$N$ model does not suffer from the "double" ABL height problem that can occur with the RANS-$\Theta$ model, because the RANS-$N$ model does not require an explicit inversion height. The RANS-$N$ model

behaves similarly to the RANS-$\ell_{\max}$ model in terms of obtaining an ABL height implicitly using a single parameter; instead of an ABL length scale arising from $\ell_{\max}$ (i.e. $z_i \simeq \ell_{\max}^{0.6}(G/f)^{0.4}$ as in van der Laan et al., 2020), the depth is determined by the constant $N_{\mathrm{ABL}}$. We note that one can also translate $N_{\mathrm{ABL}}$ to an ABL length scale using $G/N_{\mathrm{ABL}}$. The latter defines an ambient turbulence length scale above the ABL:

$$\ell_{\mathrm{amb}} = C_{\mathrm{amb}} \frac{G}{N_{\mathrm{ABL}}}, \tag{13}$$

with $C_{\mathrm{amb}} = 10^{-7}$ and $I_{\mathrm{amb}} = 10^{-5}$. If $N_{\mathrm{ABL}} = 0$, then $\varepsilon_{\mathrm{amb}}$ is set to zero. Since the RANS-$N$ model does not use the global length scale limiter of Apsley and Castro (1997) ($C_{\varepsilon,1}^* = C_{\varepsilon,1}$), the model does not artificially limit the turbulence length scale in a 3D RANS simulation. The remaining constant, $C_{\varepsilon,3}$, is set the same as the RANS-$\Theta$ model (Eq. 10).

## 2.4 Similarity

The RANS ABL models discussed here ultimately depend on four or five dimensional parameters, but their non-dimensional

numerical solutions can be described by two or three dimensionless numbers (following the Buckingham-Pi theorem), as summarized in Table 1. The first dimensionless number is the surface Rossby number, $\mathrm{Ro}_0 \equiv G/(|f_c|z_0)$, and can be obtained

| Model | Dimensional input | Non-dimensional input |
|---|---|---|
| RANS-$\ell_{\max}$ | $G, f_c, z_0, \ell_{\max}$ | $\mathrm{Ro}_0, \mathrm{Ro}_\ell$ |
| RANS-$\Theta$ | $G, f_c, z_0, z_i, \frac{d\Theta}{dz}\big|_c, \theta_0$ | $\mathrm{Ro}_0, \mathrm{Ro}_{z_i}, \mathrm{N}_f$ |
| RANS-$N$ | $G, f_c, z_0, N_{\mathrm{ABL}}$ | $\mathrm{Ro}_0, \mathrm{N}_f$ |

**Table 1.** Dimensional and non-dimensional input parameters of RANS inflow models.

by writing the 1D momentum equations (Eq. 1) in a complex form using $W \equiv (U - U_G) + i(V - V_G)$, with $i \equiv \sqrt{-1}$, followed by a substitution of the normalized variables, $z' \equiv z/z_0$, $W' \equiv W/G$ and $\nu'_T \equiv \nu_T/(z_0 G)$:

$$\mathrm{Ro}_0 \frac{d}{dz'}\left(\nu'_T \frac{dW'}{dz'}\right) = iW'. \tag{14}$$

All models that solve the momentum equation (14) follow a Rossby similarity. The other dimensionless numbers are related to the turbulence model equations (2), which can be written in a non-dimensional form using $k' = k/G^2$, $\varepsilon' = \varepsilon z_0/G^3$:

$$\frac{d}{dz'}\left(\frac{\nu_T'}{\sigma_k}\frac{dk'}{dz'}\right) + \mathcal{P}' + \mathcal{B}' - \varepsilon' = 0,$$

$$\frac{d}{dz'}\left(\frac{\nu_T'}{\sigma_\varepsilon}\frac{d\varepsilon'}{dz'}\right) + \left(C_{\varepsilon,1}^*\mathcal{P}' - C_{\varepsilon,2}\varepsilon' + C_{\varepsilon,3}\mathcal{B}'\right)\frac{\varepsilon'}{k'} = 0,$$

(15)

with $\mathcal{P}' \equiv \mathcal{P}z_0/G^3$ and $\mathcal{B}' \equiv \mathcal{B}z_0/G^3$. Here, the small ambient source terms are neglected. The additional dimensionless numbers are obtained from non-dimensionalizing either $C_{\varepsilon,1}^*$ (Eq. 5) or $\mathcal{B}'$ (via Eqns. 3, 8, 11):

$$
\begin{aligned}
&\text{RANS}-\ell_{\max}: &\mathcal{B}' &= 0, &C_{\varepsilon,1}^* &= C_{\varepsilon,1} + (C_{\varepsilon,2} - C_{\varepsilon,1})C_\mu^{3/4}\frac{k'^{3/2}}{\varepsilon'}\frac{\text{Ro}_\ell}{\text{Ro}_0}, \\
&\text{RANS}-\Theta: &\mathcal{B}' &= -\frac{\nu_T'}{\sigma_\theta}\left(\frac{N_f}{\text{Ro}_0}\right)^2\left[\frac{1}{2} + \frac{1}{2}\tanh\left(\frac{z'\text{Ro}_{z_i}/\text{Ro}_0 - 1}{z_T/z_i}\right)\right], &C_{\varepsilon,1}^* &= C_{\varepsilon,1}, \\
&\text{RANS}-N: &\mathcal{B}' &= -\frac{\nu_T'}{\sigma_\theta}\left(\frac{N_f}{\text{Ro}_0}\right)^2, &C_{\varepsilon,1}^* &= C_{\varepsilon,1},
\end{aligned}
$$

(16)

where $\text{Ro}_\ell \equiv G/(|f_c|\ell_{\max})$ and $\text{Ro}_{z_i} \equiv G/(|f_c|z_i)$ are Rossby numbers based on different ABL length scales, namely, $\ell_{\max}$ and $z_i$, respectively. In addition, $N_f \equiv N/|f_c|$ is the Zilitinkevich number using the Brunt-Väisälä frequency from Eq. (12) using a constant gradient of temperature (representing the inversion or the entire ABL for the RANS-$\Theta$ and RANS-$N$ models, respectively). Note that one could also replace $N_f$ by a Richardson number, in the form of $(N_f/\text{Ro}_0)^2$. The similarity of the RANS-$\ell_{\max}$ and RANS-$\Theta$ models has been shown through numerical experiments in previous work (van der Laan et al., 2020, 2023b). The similarity of the ABL models can be employed to create an ABL library numerically for all possible solutions, which can be used to obtain an ABL profile with a desired turbulence intensity and wind speed at a reference height by using $G$ and $N_{\text{ABL}}$ (in the case of the RANS-$N$ model) as free parameters, for a given $f_c$ and $z_0$. The proposed RANS-$N$ model has one fewer dimensional number compared to the RANS-$\Theta$ model, which reduces the input parameter space.[1] In addition, all three RANS models can be used to satisfy Reynolds number similarity by keeping their non-dimensional numbers constant. This is an advantage when running wind speed inflow cases consecutively to reduce the total number of required iterations for wind farm AEP simulations (van der Laan et al., 2019, 2022).

## 3  Numerical methodology

The RANS simulations of the inflow and single turbine wake are carried out with PyWakeEllipSys (DTU Wind and Energy Systems, 2024), which is Python framework for wind farm CFD simulations. The underlying CFD solver is EllipSys; which is an in-house finite volume code initially developed by Michelsen (1992); Sørensen (1994). The numerical domain and boundary conditions of the 1D inflow precursor and 3D wind turbine simulations are depicted in Fig. 1, and are further discussed in Sects. 3.1 and 3.2.

---

[1]It could appear that the RANS-$\Theta$ model further includes the parameter $z_T$, but this may be eliminated by relating $N_{\text{ABL}}$ to $N(z)$ and $N_c$ following Kelly et al. (2019); however this is beyond the scope of the current work.

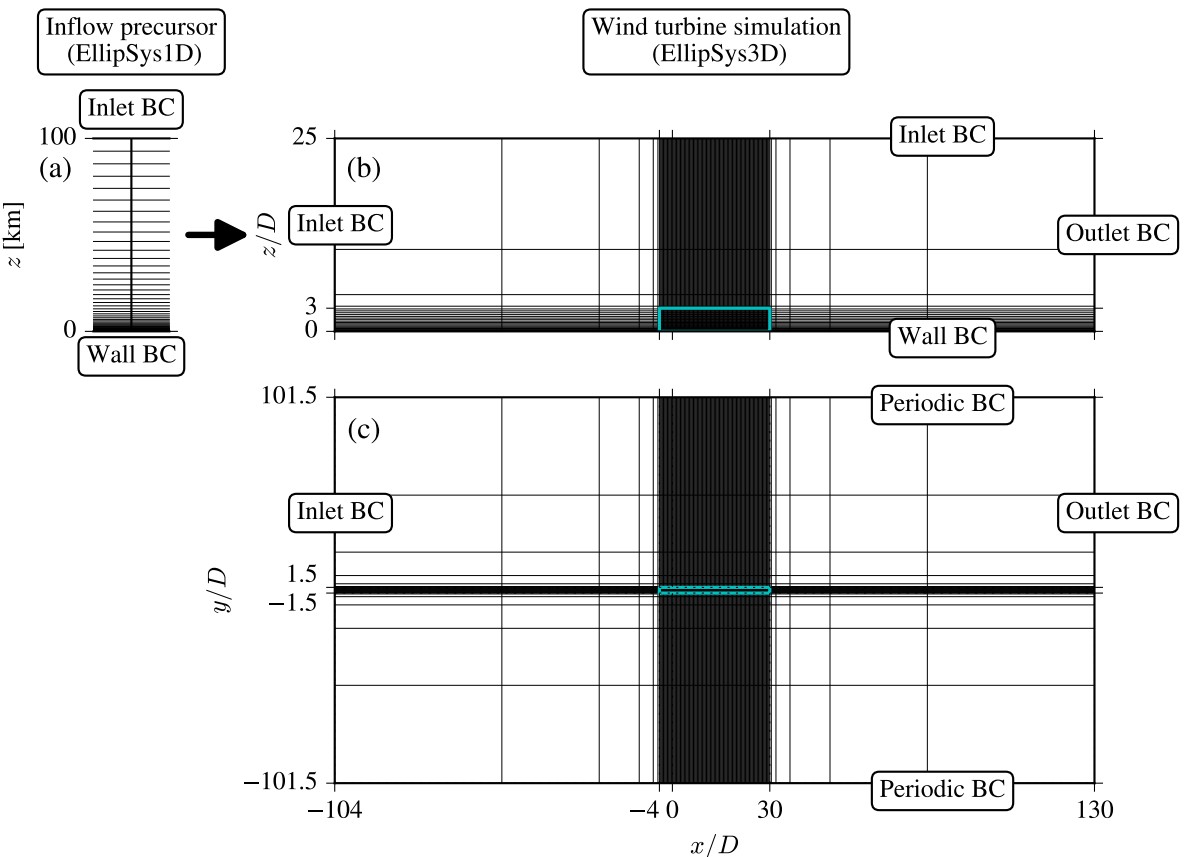

**Figure 1.** Numerical grid and boundary conditions of the 1D inflow precursor **(a)** and 3D wind turbine simulations **(b-c)**. Cyan rectangle marks the refined domain around the turbine and every 8th cells is shown.

## 3.1 Inflow

The RANS inflow models are solved numerically with EllipSys1D (van der Laan and Sørensen, 2017). A 1D grid with a height of 100 km, a first cell height of 0.01 m and 768 cells are employed, as shown in Fig. 1a. A relative tall domain is employed to be able to simulate all possible ABL solutions, as discussed in van der Laan et al. (2020). A rough wall boundary condition from Sørensen et al. (2007) is employed at the ground, and depends on the roughness length $z_0$. At the top, a Neumann condition is applied. Since the 1D RANS equations are stiff, we solve them transient with a fixed time step of $\Delta t = 1/f_c$ until a steady-state has been achieved.

## 3.2 Single wake

The RANS inflow models are applied to RANS single wake simulations, performed with EllipSys3D. The numerical setup follows a very similar approach as performed in previous work (van der Laan et al., 2015c) and solves the three-dimensional

form of Eqns. (1-2). We aim to compare the RANS simulations (both inflow and turbine wakes) with results of two LES models from Hodgson et al. (2023). These LES models employ an Actuator Disk (AD) model based on airfoil data to represent the forces of the SWT-2.3-93 turbine (propriety to Siemens Gamesa Renewable Energy), which has a rated power of 2.3 MW, a rotor diameter, $D$, of 93 m, and a hub height, $z_H$, of 68.5 m. Our RANS simulations use the same turbine type, but we employ an AD (Réthoré et al., 2014) including the analytic blade force distribution model of Sørensen et al. (2020), which has shown to compare well with an AD based on airfoil data. In order to perform a fair comparison between the LES and RANS models, we have rerun the LES wake simulations using the same AD model as applied in RANS, see Sect. 3.2.1 for more details. The AD model includes effects of rotor rotation and non-uniform inflow as wind shear and wind veer. In addition, we use a 1D momentum controller (Calaf et al., 2010) similar to one of LES models from Hodgson et al. (2023), based on the same inputs: a tip speed ratio of 7.75 and a thrust coefficient of 0.73 and a power coefficient of 0.45. Note the actual values can differ because a 1D momentum controller typically overestimates the freestream wind speed and thrust force, as shown in previous work (van der Laan et al., 2015b). The effective values of the power and thrust coefficients based on the disk-averaged streamwise velocity are set as 1.026 and 1.264, respectively.

A Cartesian domain is employed with dimensions $234D \times 203D \times 30D$ for the streamwise ($x$), lateral ($y$) and vertical ($z$) directions, respectively, as depicted in Fig. 1a-b. The large domain extend is used to minimize the effect of numerical blockage. An inner domain around the turbine, located at $(x, y, z) = (0, 0, z_H)$, is used to resolve the wind turbine wake with a fine uniform spacing of $D/32$ (cyan rectangle in Fig. 1a-b). The inner domain has the following horizontal dimensions: $-4D < x < 30D$ and $-1.5D < y < 1.5D$. Vertically, the cell sizes are growing with $z$ using a first cell height of $D/200$, a maximum cell size of $D/32$ at $z = 3D$ and a maximum expansion ratio of 1.2. Above $z = 3D$, the cells continue to grow with a similar expansion ratio. The total number of cells is 43.6 million. The effect of coarser grid spacing is shown in Appendix B. An inlet boundary condition is at the inflow boundary ($x = -104D$) and at the top of domain ($z = 25D$). The bottom boundary is a rough wall boundary condition (Sørensen et al., 2007). The lateral boundaries ($y = \pm 101.5D$) are periodic because of the presence of wind veer. A Neumann condition is set at the outflow boundary ($x = 130D$). More details of the numerical setup are discussed in van der Laan et al. (2015c) with the exception of the lateral boundaries, which are set to periodic boundary conditions because of the presence of wind veer.

### 3.2.1 LES

The LES results of Hodgson et al. (2023) are used to compare with our RANS models results for both the inflow and single wake cases. Hodgson et al. (2023) employed two different LES models; WiRE (Albertson and Parlange, 1999; Porté-Agel et al., 2000; Wu and Porté-Agel, 2011) and EllipSys3D (same solver as used for the RANS simulations), here labeled as LES-EPFL and LES-DTU, respectively. In order to provide a fair comparison with the RANS models, we have rerun the LES-DTU single wake cases following the same methodology as Hodgson et al. (2023), but using a finer grid spacing of $D/32$ around the AD instead of $D/16$. In addition, we have extended the refined domain around the AD in the streamwise direction to $30D$ for the SBL inflow case, such that we can compare LES-DTU results of the far wake with RANS. Furthermore, we have extended the precursor simulation by two additional hours such that the LES-DTU SBL single wake simulation can be averaged over

three hours in order to obtain converged statistics in the far wake. Finally, we employ the same AD model as used in the RANS
simulations including a 1D momentum controller for both inflow cases.

## 3.3 Wind turbine row with SBL inflow

The SBL inflow case is also applied to a small wind farm consisting of a row of five SWT-2.3-93 turbines (same turbine
as used for the single wake cases) with $5D$ spacing. The RANS wind farm domain is similar to the domain used for the
single wake cases (as depicted in Fig. 1). However, a larger inner domain is used with the following horizontal dimensions:
$-4D < x < 40D$ and $-3.5D < y < 3.5D$ leading to a total number of 94.4 million cells. The wind turbine row subjected to
the SBL inflow case is also simulated with the LES-DTU model using the same extended domain as used for the SBL single
wake case, as discussed in Sect. 3.2.1.

## 4 Results and discussion: A comparison with LES

### 4.1 Inflow

The two existing and proposed RANS inflow models from Sect 2 are applied to two ABL cases based on LES results from
Hodgson et al. (2023), who used two different LES models. The ABL cases represent a CNBL and a stable ABL (SBL) inspired
by the LES inter-comparison study from Beare et al. (2006). The LES models from Hodgson et al. (2023) are employed with
$f_c = 1.185 \times 10^{-4}$ s$^{-1}$ and $z_0 = 0.001$ m. The values of the Coriolis parameter corresponds to a latitude of 54.3° and it is
based on the location of the Danish offshore wind farm, Rødsand II. While we adopt the value of $f_c$ from Hodgson et al.
(2023), a lower roughness length is used in the RANS models. This is because the RANS models use $C_\mu = 0.03$ while the
LES models imply a higher effective $C_\mu$ based on the turbulent kinetic energy and friction velocity near the wall, as shown
in Baungaard et al. (2024). This is compensated by using a lower roughness length of $z_0 = 0.0002$ m in RANS-$\ell_{max}$ and
RANS-$N$ models. Note if a higher $C_\mu$ would be set in the RANS models then the other turbulence model constants need to
be adjusted and calibrated, which is not the scope of the present article. The RANS inflow models use $G$ and an additional
260 parameter to obtain the turbulence intensity based on $k$, $I_H$, and wind speed, $U_H$, at the reference height of 68.5 m, namely,
$\ell_{max}$, $z_0$ and $N_{ABL}$ for RANS-$\ell_{max}$, RANS-$\Theta$ and RANS-$N$, respectively. The LES-derived input parameters and fitted RANS
inflow model parameters are listed in Table 2. The RANS-$\ell_{max}$ and RANS-$N$ models use pre-calculated libraries of all possible
ABL solutions that depend on two non-dimensional numbers (as discussed in Sect. 2.4), to look up the values for $G$ and an
ABL scale ($\ell_{max}$ or $N_{ABL}$), for a given set of $I_H$ and $U_H$. The RANS-$\Theta$ model uses an optimizer to find the values of $G$ and $z_0$
for the CNBL case. LES-diagnosed values of $\theta_0$, $z_i$ and $d\Theta/dz|_c$ are not necessary for the SBL case because we do not employ
the RANS-$\Theta$ model for this case.

The RANS inflow model results are compared with the two LES models from Hodgson et al. (2023), for the CNBL and
SBL cases in Fig. 2. The result of the LES models compare well with the results of the RANS-$\ell_{max}$ and RANS-$N$ models in
terms of wind speed and turbulence intensity based on $k$ ($I_k = \sqrt{2/3k}/\sqrt{U^2 + V^2}$), for both ABL cases (Fig. 2a, c, g and i).

| | LES-derived input | | | | | RANS-$\ell_{max}$ | | RANS-$\Theta$ | | RANS-$N$ | |
|---|---|---|---|---|---|---|---|---|---|---|---|
| | $I_H$ | $U_H$ | $\theta_0$ | $z_i$ | $d\Theta/dz|_c$ | $G$ | $\ell_{max}$ | $G$ | $z_0$ | $G$ | $N_{ABL}$ |
| Case | [%] | [ms⁻¹] | [K] | [m] | [Km⁻¹] | [ms⁻¹] | [m] | [ms⁻¹] | [m] | [ms⁻¹] | [s⁻¹] |
| CNBL | 5.3 | 8.4 | 277.3 | 650 | $3.75 \times 10^{-3}$ | 9.67 | 30.7 | 9.31 | $9.31 \times 10^{-5}$ | 9.56 | $3.90 \times 10^{-3}$ |
| SBL | 3.1 | 8.8 | - | - | - | 9.58 | 3.38 | - | - | 9.85 | $2.71 \times 10^{-2}$ |

**Table 2.** LES-derived input from ABL cases and fitted parameters of RANS inflow models.

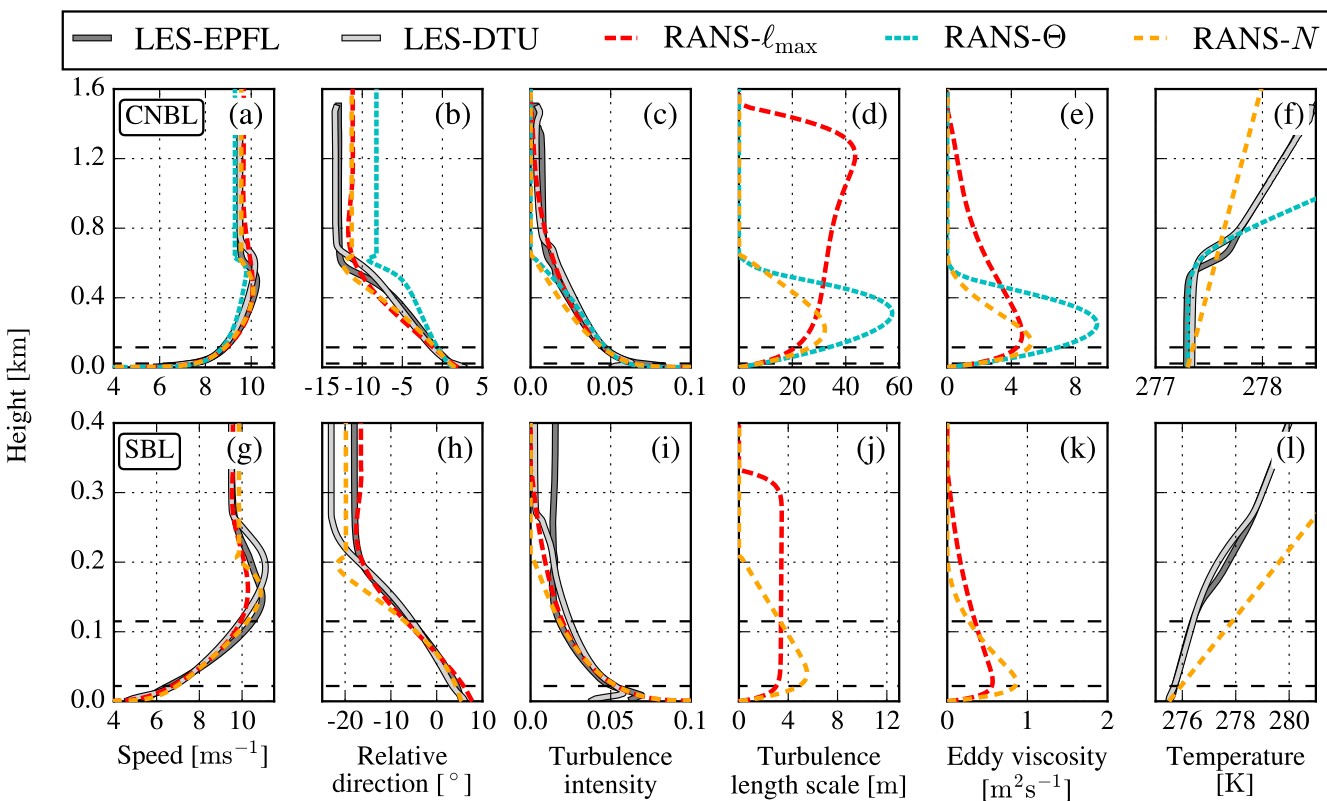

**Figure 2.** RANS-simulated ABL inflow compared to LES model results from Hodgson et al. (2023), for CNBL **(a-f)** and SBL inflow cases **(g-l)**. Horizontal dashed lines represent rotor swept area of the SWT-2.3-93 wind turbine.

The good fit around hub height is expected since the RANS models are tuned for the LES-predicted values of $I_H$ and $U_H$. The RANS predicted wind veer as shown by the relative wind directions in Fig. 2b and Fig. 2h also compares well with both LES for the CNBL case. However, the RANS-$\ell_{max}$ and RANS-$N$ models predict stronger wind veer over the rotor area (11.1° and 12.7°, respectively) compared to the LES (8.8° and 9.9° for DTU and EPFL, respectively) for the SBL case. The main difference between the RANS models are the profiles of turbulence length scale (Fig. 2d, j) and eddy viscosity (Fig. 2e and

k), where the RANS-$\ell_{\max}$ model predicts taller ABLs compared to the RANS-$N$ model; the veer difference is due in part to different effective ABL heights (Kelly and van der Laan, 2023).

Note that it is not trivial to post process an eddy viscosity or turbulence length scale from the LES data that can be directly compared to the RANS models. This is because one would need additional modeling to obtain an eddy viscosity implied by the LES data. Furthermore, the RANS turbulence length scale is a model definition, while an LES-derived turbulence length scale can be ambiguous, and is only qualitatively comparable (van der Laan and Andersen, 2018) unless non-Boussinesq contributions are accounted for (e.g. Large et al., 2019). For the SBL case, it is clear that the turbulence length scale in the RANS-$\ell_{\max}$ model is limited to a maximum value of 3.4 m, while the turbulence length scale of the RANS-$N$ model results in a more smooth profile that has a higher value in the surface layer but a lower value around the ABL height. As a result, the profiles of wind speed and direction around the ABL height are more diffused in the RANS-$\ell_{\max}$ model compared to the RANS-$N$ model, which is best visible for the SBL case (Fig. 2g and h) around $z = 0.2$ km. In other words, the RANS-$N$ has more more pronounced Ekman layer. The RANS-$N$ model predicts a lower ABL height compared to the LES models for both ABL cases, but this could be improved by lowering the applied roughness length. The latter is not performed in order to provide a more fair comparison between the RANS-$N$ and RANS-$\ell_{\max}$ models by using the same roughness length. The RANS-$\Theta$ model compares well with the LES models for the CNBL case but shows a larger turbulence length scale and eddy viscosity compared to the RANS-$N$ model due to a zero turbulent buoyancy in the surface layer (Fig. 2d and e). The RANS-$\Theta$ model is not applied to the SBL case because the RANS-$\Theta$ model cannot represent an SBL nor a shallow CNBL, as discussed in Appendix A. Results of the implied temperature profile of the RANS-$N$, $\Theta(z)/\theta_0 = 1 + zN_{\mathrm{ABL}}^2/g$, are shown in Fig. 2f and Fig. 2l. It is clear that the employed temperature gradient is larger in the RANS-$N$ with respect to the LES models, although a direct comparison with LES in terms of a temperature profile may not be fair due to the simplicity of the RANS-$N$ model. In addition, the choice of the turbulent Prandtl number in Eq. (11), here we use $\sigma_\theta = 1$, will also determine the implied temperature gradient of the RANS-$N$ model because it influences the obtained value of $N_{\mathrm{ABL}}$. One could match a constant temperature gradient to the LES results, however, it is not guaranteed that the RANS-$N$ model will compare well with the LES results in terms of wind speed, direction and turbulent intensity profiles.

## 4.2 Single wake

The RANS inflow models are applied to single turbine wake simulations and the results of velocity deficit magnitude and wake added turbulence intensity are compared with results from two LES models of Hodgson et al. (2023) in Fig. 3. The wake results are normalized by the simulation results without a turbine. The RANS-$\Theta$ model is only applied to the CNBL case and not the SBL case because the model cannot represent a shallow ABL, as discussed in Appendix A. The CNBL case shows that all three RANS inflow models predict similar velocity deficits that follow the trends of the LES models (Fig. 3b-d). The differences between the RANS and LES models in the near wake at $x = 1D$ (Fig. 3a) are expected, following a previous study (van der Laan et al., 2015c). The difference in velocity deficit between the RANS and LES models for the SBL case are larger than the CNBL case. The largest difference between the RANS-$\ell_{\max}$ and RANS-$N$ models is observed at the far wake at $x = 25D$, for the SBL case, which is depicted in Fig. 4. Figure 4a shows that the RANS-$\ell_{\max}$ model does not allow the

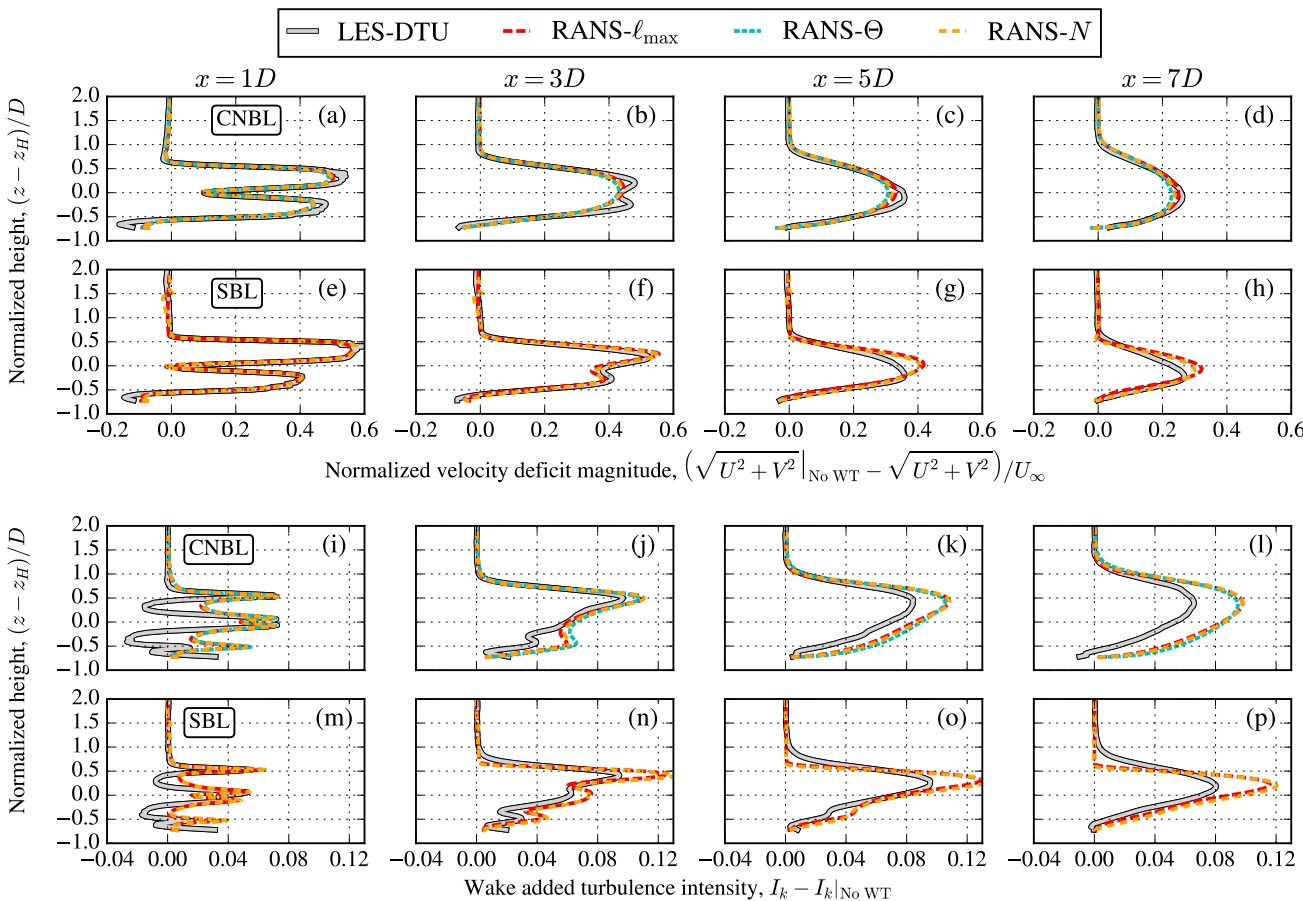

**Figure 3.** RANS-simulated wake velocity deficit **(a-h)** and wake added turbulence intensity **(i-p)** compared to LES model results, for the CNBL **(a-d, i-l)** and SBL inflow cases **(e-h, m-p)**. LES wake added turbulence intensity includes resolved and sub-grid model results.

turbine wake to recover vertically with respect to the RANS-$N$ model due to the global length scale limiter. The LES results at $x = 25D$ suggest that RANS-$N$ model better predicts the vertical wake recovery, although the magnitude of the velocity deficit is slightly better captured by the RANS-$\ell_{\max}$. A future study is needed to further validate the results of RANS-$N$ model for additional LES cases that differ in atmospheric conditions.

The RANS-$N$ model results of the SBL single wake case shows a small speed up around the ABL height $(z - z_H)/D \approx 1.4$ at $x = 1D$ (Fig. 3e), which grows further downstream (Fig. 3f-h and Fig. 4a). This is a numerical issue associated with the low eddy viscosity at the ABL height that can also occur with the other RANS inflow models, especially when they are applied to a large wind farm (van der Laan et al., 2023b). A possible solution is an additional damping method in the momentum equation. Since the proposed RANS-$N$ model does not limit the turbulence length scale globally, one could add a high eddy-viscosity damping layer above the ABL through additional sources of $k$ and $\varepsilon$ in the transport equations. Such a damping layer

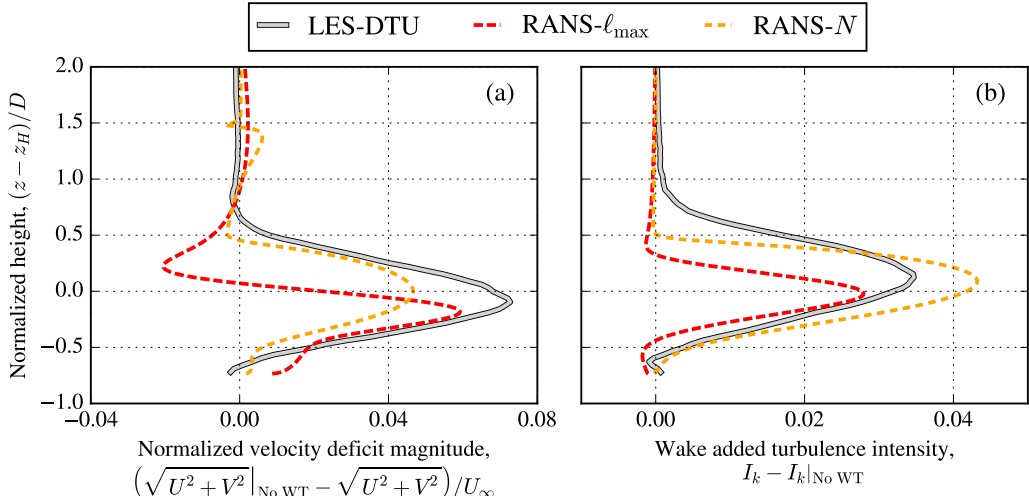

**Figure 4.** RANS-simulated wake velocity deficit **(a)** and wake added turbulence intensity **(b)** compared to LES-DTU model results, for the far wake at $x = 25D$ of the SBL inflow case. LES wake added turbulence intensity includes resolved and sub-grid model results.

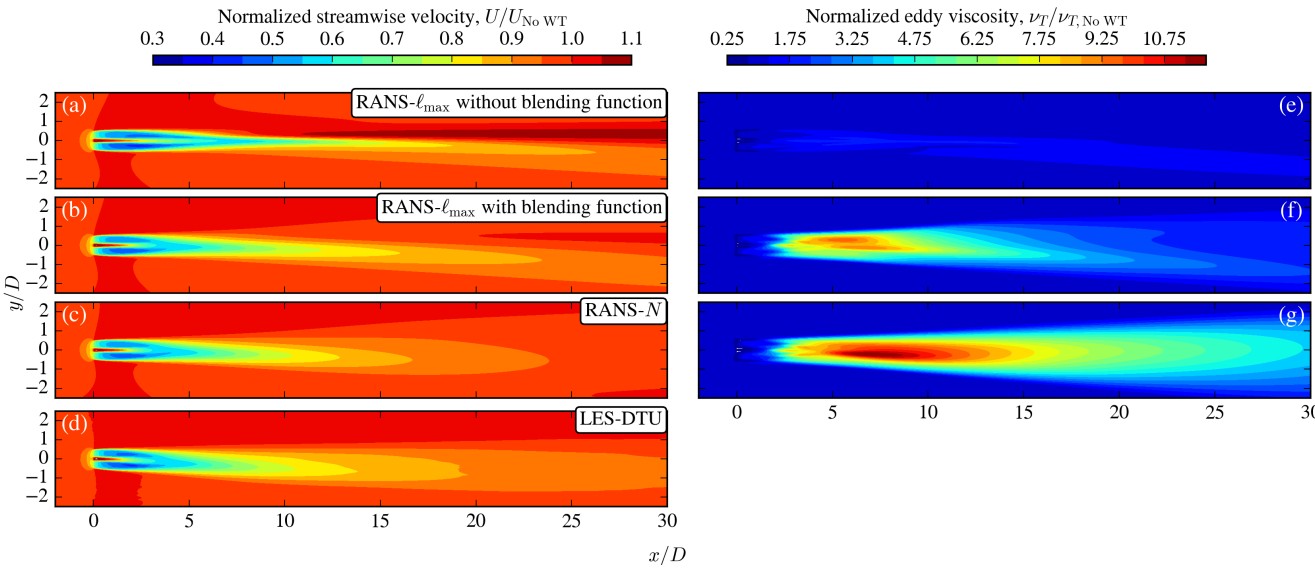

**Figure 5.** Hub height contours of normalized streamwise velocity **(a-d)** and wake eddy viscosity normalized by inflow eddy viscosity **(e-f)** for SBL case. RANS-$\ell_{\max}$ model without blending function of Eq. (6) **(a, e)**, RANS-$\ell_{\max}$ model with blending function of Eq. (6) **(b, f)**, RANS-$N$ model **(c, g)**, and LES-DTU model **(d)**.

can presumably be made to not influence the inflow profiles, while still damping the numerical 'wiggles' in the wind turbine
simulation. The required amount of damping is case-dependent and needs further study, which is part of ongoing work.

None of the RANS models are able to predict the wake added turbulence intensity compared to LES (Fig. 3i-p) because
of the applied isotropic Boussinesq hypothesis. However, the RANS models do not need a good prediction of wake added
turbulence intensity in order to predict a realistic velocity deficit because the wake recovery is dictated by the divergence of the
shear stresses (van der Laan et al., 2023a); the latter can be well modeled by the isotropic Boussinesq hypothesis and a variable
$C_\mu$ (for example through $f_P$),

Contours of the steamwise velocity and eddy viscosity at hub height of the SBL single wake case are shown in Fig. 5 for the
RANS-$N$ and RANS-$\ell_{\max}$ models. Two results of the RANS-$\ell_{\max}$ model are shown, one with and one without the blending
function of Eq. (6) that is used to switch off the global turbulence length scale limiter in the wake region (identified by the
$f_P$ function). Without Eq. (6), the eddy viscosity does not increase significantly because of the global turbulence length scale
limiter (Fig. 5e), which delays the recovery of the streamwise velocity deficit and shows a speed up region in far wake (Fig. 5a).
When Eq. (6) is included, the eddy viscosity can increase downstream but it quickly returns to the ambient eddy viscosity in the
region where the $f_P$ is close to one (Fig. 5f). The RANS-$N$ model does not limit the turbulence length scale as the RANS-$\ell_{\max}$
model, which results in a smoothly increasing (up to $x = 10D$) and decreasing eddy viscosity (Fig. 5g). As a result, a smoother
far-wake velocity deficit is obtained by the RANS-$N$ model (Fig. 5c), which explains the difference between the RANS-$\ell_{\max}$
and RANS-$N$ models at $x = 25D$ shown in Fig. 3j. In addition, the streamwise velocity of the LES-DTU model (Fig. 5d)
compares better with the results of the RANS-$N$ than the RANS-$\ell_{\max}$ up to a distance of about $20D$ downstream. Further
downstream, the RANS-$N$ predicts less velocity deficit at hub height compared to LES, as shown previously in Fig. 4a.

### 4.3  Wind turbine row with SBL inflow

Simulation results of a wind turbine row consisting of five turbines with $5D$ spacing in the streamwise direction, subjected
to the SBL inflow, are depicted in Figs. 6–8. Contours of normalized streamwise velocity are shown in Fig. 6a-d and Fig. 7,
at hub height and at five cross planes, respectively, for the RANS and LES-DTU models. Two results of the RANS-$\ell_{\max}$ are
shown, similar to the single-wake results in Fig. 5. Without the blending function, the wake recovery inside the wind turbine
row is slow (Fig. 6a) because the eddy viscosity is not growing downstream due to the global length scale limiter (Fig. 6e).
In addition, the global length scale limiter also affects the vertical wake recovery leading to wake shapes (Fig. 7b-e) that do
not resemble the LES results at all (Fig. 7q-t). When the blending function is used (Fig. 6b), the wakes of the first turbines are
more comparable with the LES results (Fig. 6d), however, further downstream the wake recovery is again too slow because the
blending function is less active at this distance, resulting in a low eddy viscosity (Fig. 6f) and artificial wake shapes (Fig. 7h-j).
The RANS-$N$ model predicts an eddy viscosity that is smooth and grows with downstream distance (Fig. 6g). As a result, the
wake recovery of the RANS-$N$ model (Fig. 6c) is faster than the RANS-$\ell_{\max}$ (Fig. 6b) and the wake shapes of the RANS-$N$
model (Fig. 7l-o) more closely resemble the LES results (Fig. 7q-t). However, the magnitude of the streamwise velocity deficit
predicted by the RANS-$N$ model (Fig. 6c) is underpredicted compared to the results of the LES-DTU model (Fig. 6d). In
addition, the wakes of the LES-DTU simulations are more deflected compared to the wakes of the RANS-$N$ model.

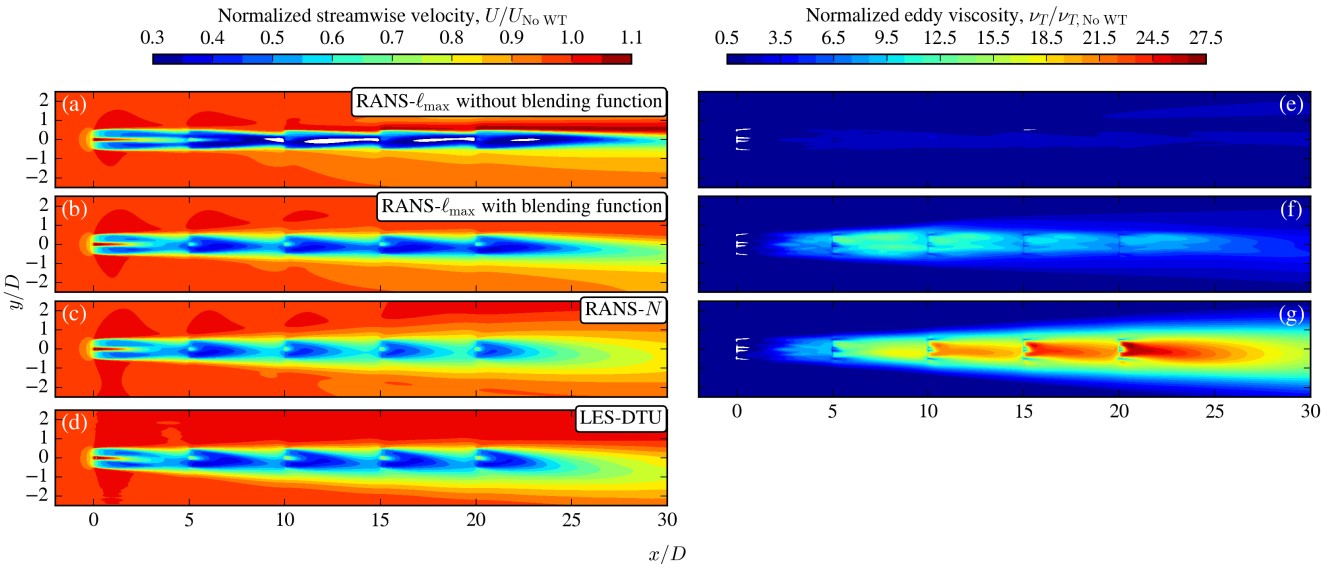

**Figure 6.** Hub height contours of normalized streamwise velocity **(a-d)** and wake eddy viscosity normalized by inflow eddy viscosity **(e-f)** for SBL case applied to a wind turbine row. RANS-$\ell_{max}$ model without blending function of Eq. (6) **(a, e)**, RANS-$\ell_{max}$ model with blending function of Eq. (6) **(b, f)**, RANS-$N$ model **(c, g)**, and LES-DTU model **(d)**.

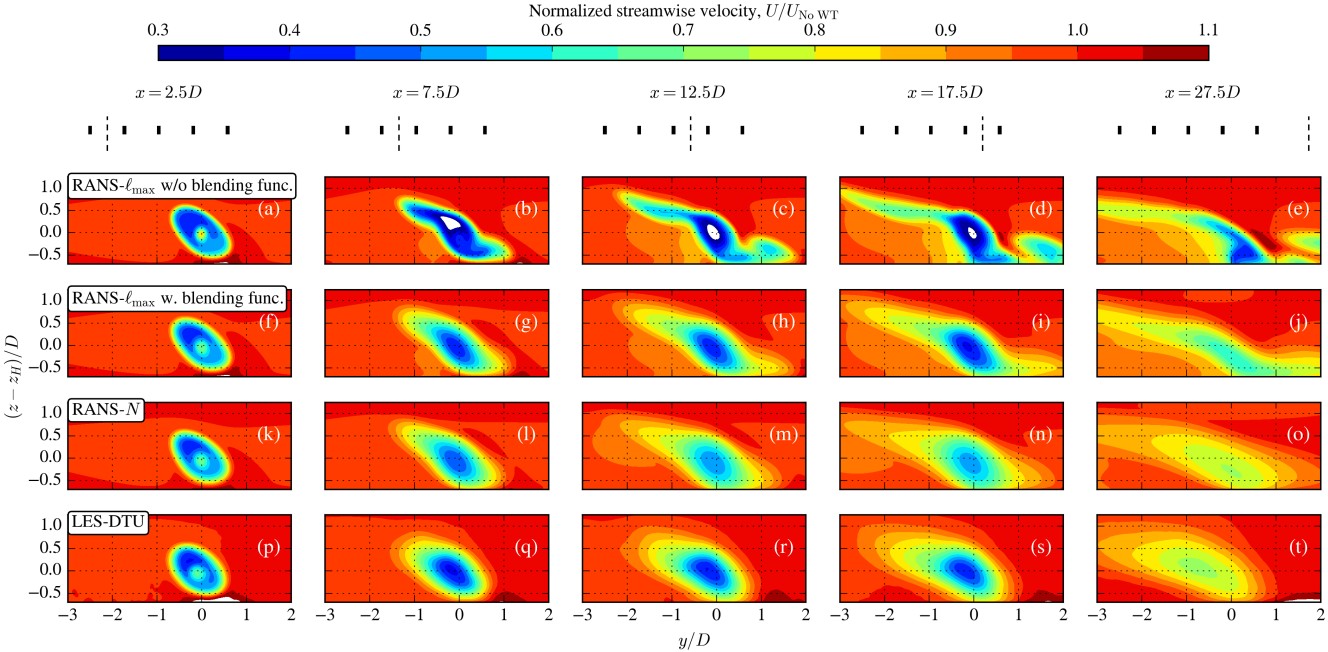

**Figure 7.** Cross-plane contours of normalized streamwise velocity for SBL case applied to a wind turbine row. RANS-$\ell_{max}$ model without blending function of Eq. (6) **(a-e)**, RANS-$\ell_{max}$ model with blending function of Eq. (6) **(f-j)**, RANS-$N$ model **(k-o)**, and LES-DTU model **(p-t)**.

Figure 8 depicts results of the streamwise velocity deficit and wake added turbulence intensity, as rotor-averaged values along the turbine row. The RANS-$\ell_{\max}$ model performs best inside the wind turbine row, while the RANS-$N$ model performs better behind the last turbine, in terms of matching the velocity deficit from the LES-DTU; this is shown in Fig. 8a. It should be noted that the LES-DTU model predicts a stronger wake deflection compared to the RANS models (as discussed previously with Fig. 6), which can affect the comparison in terms of rotor-averaged results. In terms of wake added turbulence intensity (Fig. 8b), neither RANS models can predict the LES results. It is also clear that the DTU-LES model predicts a loss in wake added turbulence intensity at the actuator disk locations. Such behavior has also been observed in other LES-AD simulation results (Abkar and Porté-Agel, 2015; García-Santiago et al., 2024). Zehtabiyan-Rezaie and Abkar (2024) proposed an additional sink of $k$ in a RANS-AD model in order to mimic the LES-AD results (without the use of an $f_P$ function). The proposed RANS-$N$ model could potentially be extended with a similar additional sink of $k$, although it is unclear if a reduction of turbulent kinetic energy at the rotor is a real phenomena or a model artifact (e.g. related to representing a rotor as an AD where blade-resolved turbulence is absent). Alternatively, the $f_P$ function (Eq. 2) could be recalibrated for stable conditions such that the diffusivity of the RANS-$N$ model is reduced. A similar exercise was performed in a previous work to model a wind turbine wake under unstable surface layer conditions (Baungaard et al., 2022a). A further development of the RANS-$N$ model requires a range of inflow cases applied to wind farms using LES. In addition, the RANS-$N$ model needs to be validated with field measurements.

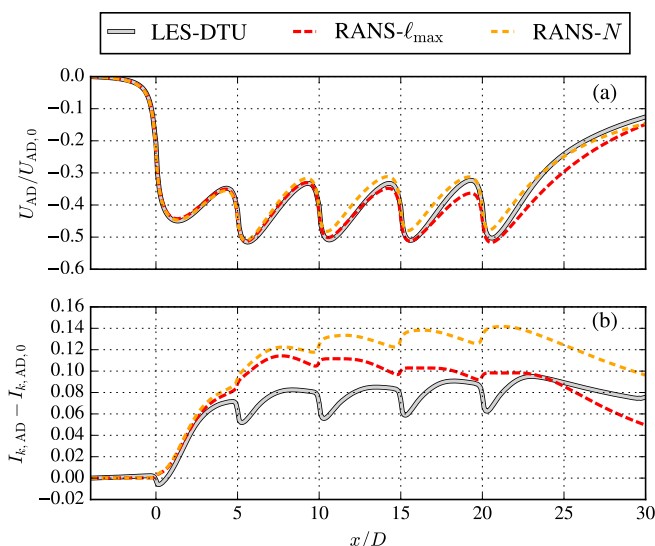

**Figure 8.** Rotor-averaged streamwise velocity deficit **(a)** and wake added turbulence intensity **(b)** normalized by their respective values at $x = -4D$, for the SBL case, applied to a wind turbine row, simulated by RANS and LES-DTU models. LES wake added turbulence intensity includes resolved and sub-grid model results.

## 5 Conclusions

A new RANS inflow model of the neutral and stable ABL is proposed and compared with two existing RANS inflow models for CNBL and SBL cases based on LES model results. The proposed inflow (RANS-$N$) model does not require a global length scale limiter or prior knowledge of temperature profile by the use of simple turbulent buoyancy expression based on a constant Brunt-Väisälä frequency. The RANS-$N$ model compares well with LES-predicted profiles of wind speed, wind direction and turbulence intensity. The simplicity of the RANS-$N$ model results in a reduced parameter space consisting of only two non-

dimensional numbers, the surface Rossby number and the Zilitinkevich number. The three RANS inflow models are applied to single wind turbine wakes for the same ABL cases and their simulated velocity deficit compares well with results from LES for the CNBL case. In addition, the SBL inflow case is applied to an along-wind wind turbine row. The present study has shown that the proposed RANS-$N$ model is better suited to simulate the effect of a shallow SBL on a single wind turbine wake and a wind turbine row, than the existing state-of-the-art RANS-$\ell_{\max}$ (Apsley and Castro, 1997) and RANS-$\Theta$ (van der

Laan et al., 2023b) models in terms of the velocity deficit shape. However, the RANS-$N$ model underpredicts the magnitude of the velocity deficit of the wind turbine row with respect to LES and further model investigation is required. In addition, the interaction of a shallow ABL and a turbine wake in RANS can lead to small numerical wiggles, which grow with downstream distance and needs further investigation for the application of large wind farm simulations.

## Appendix A:  Caveat regarding use of RANS-$\Theta$ inflow model with $k$-$\epsilon$ closure

The RANS-$\Theta$ model can predict a 'double' ABL height, if a strange combination of the input parameters is chosen; this happens if the $k$-$\epsilon$ model implies a value of $z_i$ significantly different from that chosen in the temperature profile $\Theta(z)$. For example, for a shallow ABL one could set a low $z_i$, but if the chosen inversion strength is not strong enough then the effective ABL height can occur above the inversion height, $z > z_i$. An example of this issue is shown in Fig. A1, where the RANS-$\Theta$ model is employed for $\{G, f_c, z_0, z_i, d\Theta/dz|_c\} = \{10\,\text{m}, 10^{-4}\,\text{s}^{-1}, 10^{-4}\,\text{m}, 100\,\text{m}, 0.1\,\text{Km}^{-1}\}$ and three different combinations

of $z_T/z_i$. The earlier prescribed value ($z_T/z_i = 0.2$) can result in the double height problem, which creates an inflection point in the wind speed profile (Fig. A1a) at $z \approx 80\,\text{m}$. When the smoothing is increased by setting larger values of $z_T/z_i$, then the double ABL height is less visible and the model behaves more like the RANS-$N$ model since the temperature gradient approaches a constant value.

One could extend the RANS-$\Theta$ model by adding a surface-layer temperature gradient, $d\Theta/dz|_s$:

$$395 \quad \frac{d\Theta}{dz} = \frac{1}{2}\left[1 + \tanh\left(\frac{1 - z/z_i}{z_T/z_i}\right)\right]\frac{d\Theta}{dz}\bigg|_s + \frac{1}{2}\left[1 + \tanh\left(\frac{z/z_i - 1}{z_T/z_i}\right)\right]\frac{d\Theta}{dz}\bigg|_c, \tag{A1}$$

which can reduce the problem with double ABL heights for a shallow and stable ABL using a positive surface layer gradient. However, the user can still obtain a double ABL height if $d\Theta/dz|_s$ is not strong enough, and for a too strong $d\Theta/dz|_s$, the model can produce a lower ABL height than intended. One could also employ a prescribed temperature gradient profile from a higher fidelity model as LES; however, a smooth wind speed speed profile is not guaranteed, research is ongoing on how to

400 ensure such. In the present work, we do not use Eq. (A1) but rather adapt the original formulation (van der Laan et al., 2023b).

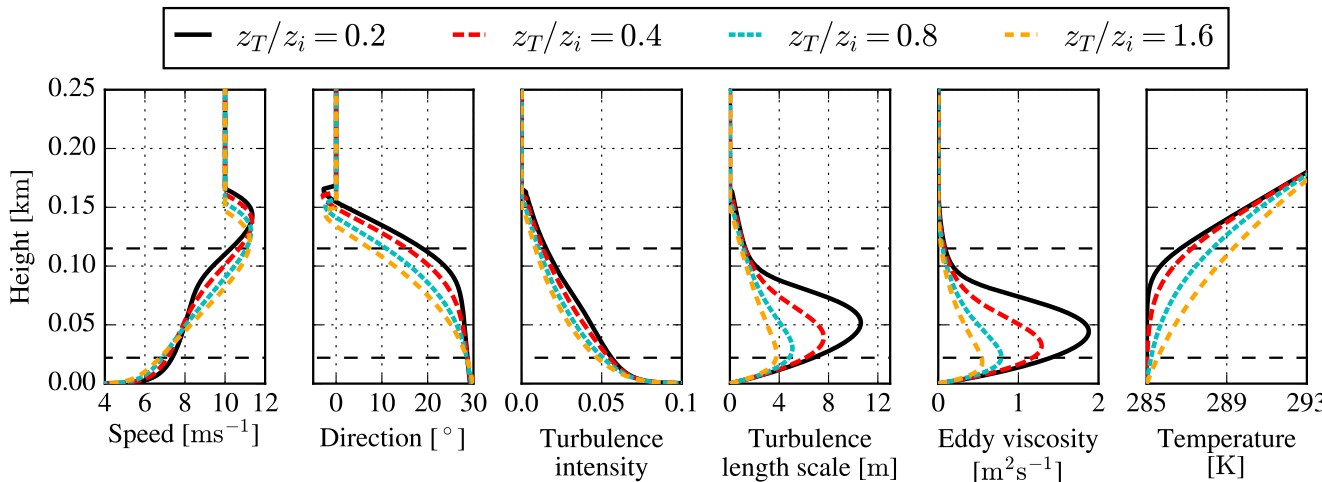

**Figure A1.** ABL inflow simulated with the RANS-$\Theta$ model for different values $z_T/z_i$. Horizontal dashed lines represent rotor swept area of the SWT-2.3-93 wind turbine.

## Appendix B: Grid refinement study of the RANS-$N$ inflow model applied to the SBL single wake case

The grid sensitivity of the proposed RANS-$N$ inflow model applied to the single wake SBL case is depicted in Fig. B1. Three coarser grids are employed compared to the results presented in the main body of the article, which leads to four different grid sizes in the domain around the turbine, $\Delta$: $D/4$, $D/8$, $D/16$ and $D/32$, which correspond to total cell counts of 0.786, 3.24, 9.96 and 43.6 million, respectively. Figure B1a shows the rotor integrated streamwise velocity normalized by the freestream and also includes a Richardson Extrapolated (RE) value following the mixed order grid convergence analysis from Roy (2003). The corresponding discretization error (normalized by the freestream velocity) is plotted in Fig. B1b and indicates that a grid spacing of $D/8$ results in an error less than 1%, at a downstream distance of $8.5D$ and beyond. Such an error is acceptable for the application of RANS wind farm simulations of modern offshore wind farms, where the typical turbine inter spacing is around $7-10D$. The rotor integrated wake added turbulence intensity and corresponding discretization error are plotted in Fig. B1c and Fig. B1d, respectively. The errors in wake added turbulence intensity are of similar magnitude with as the error in velocity deficit. A grid spacing of $D/8$ results in an error of about 0.5% at a downstream distance of $7.5D$.

*Code and data availability.* The numerical results are generated with proprietary software, although the data presented can be made available by contacting the corresponding author.

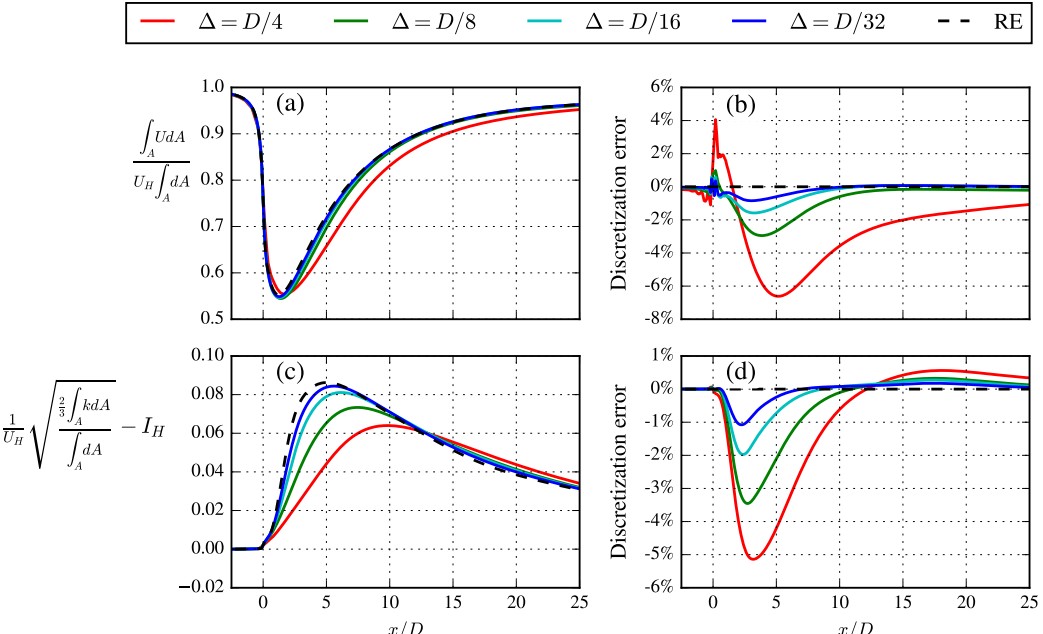

**Figure B1.** RANS wake streamwise velocity **(a)** and wake added turbulence intensity **(c)**, integrated over a fictitious rotor area and corresponding discretization error **(b,d)** simulated by the RANS-$N$ model, for different grid resolutions, for the SBL inflow case.

*Author contributions.* MPVDL has developed the new ABL inflow model, drafted the article and produced the figures. AD has improved the numerical stability of the numerical simulations. ELH has designed the main LES setup. All authors contributed to discussion of the new model, the methodology and finalization of the paper.

*Competing interests.* The authors declare that they have no conflict of interest.

*Financial support.* This work has been partially supported by the MERIDIONAL project, which receives funding from the European Union's
Horizon Europe Programme under the grant agreement No. 101084216. In addition, this work has also been co-financed by Equinor ASA.

*Acknowledgements.* We would like to thank Fernando Porté-Agel and Marwa Souaiby for providing their LES precursor results. We also gratefully acknowledge the computational and data resources provided on the Sophia HPC Cluster at the Technical University of Denmark, DOI: 10.57940/FAFC-6M81.

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
