# Peer review of "A simple RANS inflow model of the neutral and stable atmospheric boundary layer applied to wind turbine wake simulations"

_Wind Energy Science, 2024_

## Author Comment (AC1)

**Reply to reviewers**

July 6, 2024

We would like to thank the two reviewers for their detailed feedback and suggestions to improve the article. In the next sections, the reviewers comments are copied and answered per comment (blue color). An additional document is provided that highlights all modifications with respect to the initial submitted version. Furthermore, we also have made the following changes:

1. We have rerun the two (DTU) LES single wakes because the single wake cases from Hodgson et al. (2023) were performed on a relative coarse grid (cells size of $D/16$ for resolving the wake), which we had misunderstood. Our own LES runs are the same as Hodgson et al. (2023) but we use a finer grid size of $D/32$. In addition, we have extended the refined domain around the turbine in streamwise direction up to 30D for the SBL case in order compare LES results of the far wake with the RANS results and we run for 4 hours (instead of 3 hours) in order to obtain converged statistics of the far wake by taking an average of the last 3 hours. This also meant that we had to extend the LES SBL precursor by two hours. Finally, we used the same actuator disk model in the LES as used in the RANS simulations. The need for rerunning the single wake LES also means that it no longer makes sense to compare with the LES results from EPFL as published in Hodgson et al. (2023). The rerun DTU LES results are shown in Figs 3, 4 and 5. Fig. 4 is a new figure comparing the RANS results with LES at the far wake ($x = 25D$). The changes to the LES runs are described in Sect. 3.2.1.

2. We have added Emily Hodgson as a coauthor, since we have used her LES setup to rerun the LES cases.

3. The results in Fig. 3 are now normalized with a simulation without a turbine, which is mainly important for the newly added LES results because its inflow is slowly developing over time (and distance).

4. We added a wind turbine row case using the SBL inflow and compared results of LES with the results of the RANS models in a new Section (Sect. 4.3). The comparison shows that the proposed RANS-$N$ model performs better than the other RANS models in terms of overall wake shape. However, the LES-predicted wake deficit magnitude is underpredicted by the RANS-$N$ model, which requires further development. We have added a discussion and updated the conclusion and abstract.

**Reviewer 1**

Thank you very much for all the effort made in this innovative work. I find the continuation and improvement that you have made in the ABL+wakes simulation in RANS very important, since high resolution LES is still very slow and expensive when you want to include the interaction of several wind farms. In my opinion, the structure of the work is very good, but it is necessary to add some case studies and results to be able to reinforce the statements that are arrived when comparing the different RANS methodologies. Below, I leave the revisions.

**Major revisions**

1. Section 4.2 Single wake: The numerical issue on top of the wind turbine for the RANS-N case in stable conditions is the major limitation of this new method to be further used in ABL+wakes simulations. The author points that this is due to the low eddy viscosity near the top of the ABL height. Also,

the eddy viscosity profile is one of the major differences between RANS-lmax and RANS-N. I think that this big issue should be clarified in this current work. For example by plotting the eddy viscosity profile from LES and deciding with RANS approach resolves better the eddy viscosity. If the LES data is not available, at least a small literature review paragraph on this topic should be made.

The reviewer points out an important issue that we are currently working on. It should be noted that the numerical problems with low ABL heights in RANS is a general issue that all three RANS models presented in this work can suffer from (not only the proposed model), see for example a previous work van der Laan et al. (2023). However, the proposed RANS-$N$ model does provide new opportunities to solve this issue with respect to the RANS-$\ell_{\max}$ model. This is because the RANS-$N$ model does not limit the eddy viscosity everywhere as the RANS-$\ell_{\max}$ model does, which means that one could add a high eddy viscosity damping layer above the ABL in the RANS-$N$ model. Such a method is currently tested and validated, and deserves a dedicated article. We have added a clarification in Sect. 4.2.

A comparison of LES and RANS in terms of eddy viscosity is not trivial because there is no unique objective way to calculate an eddy viscosity in LES, due to LES accounting for terms in the resolved stress rate equations which go beyond the assumption of an eddy viscosity (e.g., Wyngaard, 2004). Further, it requires an additional modeling method to calculate an eddy viscosity by for example solving an $\varepsilon$ equation with the LES flow as input. One could also compare the model turbulence length scale of RANS with a diagnosed length scale of LES (auto correlation or integrated spectra) but such comparisons are not trivial, see for example van der Laan and Andersen (2018). Therefore, we prefer to not do such a comparison in the present work. We have added a discussion in the Sect. 4.1.

2. Section 4.2 Single wake: Figure 4 shows that the largest differences between the RANS methods are located at downstream distances larger than 7D (both for velocity deficit and eddy viscosity). Since there is no more LES data to compare with, the authors could propose another sub section where, for example, the power output of a small wind farm test is compared using RANS, LES and, if available, SCADA.

   We agree that we need LES data further downstream to be able to confirm which model performs best. Therefore, we have rerun the SBL LES case with an extended domain up to 30D downstream and using a 3 hour single wake average as apposed to a single hour in order to obtain converged statistics of the far wake. LES results at 25D are now compared with RANS in Fig. 4, which shows that the RANS-$N$ model allows the wake to diffuse vertically that is comparable with LES. The RANS-$\ell_{\max}$ model limits this vertical diffusion artificially due to the global length scale limiter. On the other hand, the wake magnitude is slightly better captured by the RANS-$\ell_{\max}$ model. We have also added a wind turbine row case using the SBL inflow. This shows that the RANS-$N$ better captures the wake shape compared to LES, although the wake magnitude is underpredicted, which requires further model developments.

3. Conclusions: The final conclusions that RANS-N is better for simulating low ABL+wakes is a rushed conclusion, especially since it triggers numerical error on top of the turbine. The comparison against LES up to 7D is not helping much in the decision. It is preferable to change that sentence, and highlight that this the beginning of the study of a new way of simulation ABL+wakes.

   We would like to refer to the previous answer regarding the performance of the RANS-$N$ model. The numerical wiggle in RANS-$N$ model is general problem with all RANS ABL models when a shallow ABL is used as an inflow for a turbine or farm, as discussed in the first answer.

**Minor revisions**

1. Abstract: Mention at the end that there is still work on how to simulate wakes in low ABL.

   We have added a sentence in the abstract to clarify this.

2. Introduction: Better mention "state of the art N-S LES implemented in CPU". It could be more complete if the authors mention papers where the temperature equation has been actively used in RANS to resolve the ABL+wakes.

   It is not clear to the authors if an efficient LES code based on CPUs is actually faster than an LES code based on GPUs. This had been ongoing discussion in the wind energy community. We do acknowledge that Lattice Boltzmann-based LES methods can be very fast on GPUs, although one needs small time

steps. Regarding the inclusion of a temperature equation in RANS we have added some literature. Note that a RANS ABL model with a temperature equation becomes unsteady RANS (URANS) as the ABL height will keep growing over time.

3. Section 2.3: Since this is the new proposed method, it is important to present first the literature review on how this equations and ideas have been previously used by other authors. If it is totally new, then the authors should also remark this as a novelty.
   The formulation of the buoyancy and the use of an $N$ parameter characterizing the stability of the entire ABL is new as far we know. We have modified the first sentence of Section 2.3 to reflect this better.

4. Section 3.1: The use of a precursor domain height of 100km looks quite new (and then used for a 2.3km 3D domain height). How the authors arrived to that decision? Is it possible to present a sensitive study on the precursor domain height?
   The reason for this very large domain height is its application to create a library of all possible ABL solutions from which a desired ABL solution is chosen as inflow for a particular single wake case. The 100 km domain height was required for a previously developed model including unstable conditions van der Laan et al. (2020), which could result in very large ABL heights. However, for the present model the domain height could probably be lowered but this is not important. We have added a reference in the article.

5. Section 4.1: How is the turbulence intensity calculated from the resulting TKE in RANS? Specially for the stable case.
   The turbulence intensity is calculated as $\sqrt{2/3k}/\sqrt{U^2 + V^2}$. We have add this to the article

6. Conclusions: It is important to mention if there is a proper way of simulating unstable conditions with any of the 3 ABL methods. Unstable conditions are the most frequently condition in the North Sea and is important to know how to correctly simulate wind farm wakes in this condition.
   We added more information in the introduction regarding the difficulty of modeling convective ABLs with steady-state models as RANS.

7. Appendix B: Could you also plot the TKE for the different mesh resolutions? That one is even more sensitive to the mesh than the velocity.
   The reviewer is right about this. We have added the wake added TI solutions and errors per grid size. Although the TI errors are found to be similar as the error in velocity deficit. It also depends how the errors are normalized. Here, we chosen to use the freestream velocity.

**Reviewer 2**

The paper concerns the development of a RANS model correction of the k-epsilon model for ABLs in conventionally neutral and stable conditions. The model consists of an added buoyancy turbulence production term, modifying the k and epsilon equations, based on an assumed constant temperature gradient in the ABL. The model is sold as (a) having a simple form, (b) requiring exactly the correct number of non-dimensional numbers for specification, and (c) reproducing well LES simulations of identical flows.

The paper is clearly written and structured, and generally I can recommend for publication, with the following revisions (mostly concerning clarification of the work):

**Major comments**

1. The authors' starting point is their own RANS-Theta model, and the model proposed here (RANS-N) is quite a minor variation on that basic idea (differing only in the prescribed potential temperature profile). My interpretation of the RANS-Theta profile is that is corresponds well to the physically common atmospheric condition of a distinct capping inversion layer, whereas the RANS-N model represents (at best) a condition found rarely in nature. This leads to some questions:

(a) The only argument the authors given against the RANS-Theta model is the fact that if "inconsistent" parameters are specified, then the flow profile is nonphysical. Then I would say: don't specify inconsistent parameters! This is true of almost any model (including the k-epsilon model itself). Please explain in detail why you consider this a downside of the model. Is it difficult to specify consistent parameters? Have the authors made efforts in this direction, and what are the conclusions?

We understand the reviewer's point. It turns out to be difficult to specify appropriate parameters for the RANS-$\Theta$ model apriori, especially for shallow ABL heights. This is because the RANS ABL height for a converged solution is implicit and may not be same as the intended ABL height set by the fixed temperature profile. In theory, one could start with an initial temperature profile and run an unsteady RANS (URANS) simulation with a temperature equation until a quasi steady-state has been achieved. Then one could fit the obtained temperature profile and run the RANS-$\Theta$ model. However, this is out of the scope of the present article and is a subject of ongoing research; here we seek simple (steady-state) solutions to include the effect of the ABL on a wind turbine wake. In addition, the RANS-$N$ is a simpler model with respect to the RANS-$\Theta$ model because only two non-dimensional parameters are needed instead of three.

(b) This reader would greatly appreciate details of the thought- and research-process that lead to the development of the RANS-Theta to RANS-N. The authors hint at their reasoning, but in particular I'm curious what lead to the specification of the linear temperature profile. I think it's not motivated by physics per se; is it just the simplest one-parameter curve that was chosen?

There is no direct physical reasoning for using a constant temperature gradient for the entire ABL. In fact, is it not always realistic as stated in Sect. 2.3. However, the use of a constant temperature gradient leads to a well behaving RANS ABL model that can produce similar inflow profiles of wind speed and direction compared to the state-of-art RANS-$\ell_{\max}$, without the problems associated with the use of a global length scale limiter for wind farm flows. This became clear when we were trying to extend the RANS-$\Theta$ model by including an additional temperature gradient in the surface layer. It turned out that this surface layer gradient, if set strong enough, could determine the ABL height and the temperature gradient of the inversion height would simply be ignored by the $k$-$\varepsilon$ model. This shows again that is very difficult to impose a proper temperature profile for the RANS-$\Theta$ model.

(c) The RANS-Theta and -N models lead to different turbulence intensity profiles and (dramatically different) turbulence length-scale and direction profiles (Figure 2). Which of these correspond better to the LES results (not plotted)? Which of these correspond better to ABLs in nature? Especially wind-turning should have a dramatic effect on wind-farm predictions, should it not? Which model is better justified in this sense?

The turbulence intensity profiles are actually quite similar between the RANS models, as shown in Fig. 2. We have received the LES inflow precursor results from Hodgson et al. (2023) and the EPFL group Wu and Porté-Agel (2011), and we added this in Fig. 2 such that we can also compare the wind direction with RANS. The comparison with LES in terms of wind direction is reasonable, although a stronger wind veer over the rotor area is obtained with the RANS models. We have added a discussion about this in Sect. 4.1 The turbulence length scale and eddy viscosity profiles are indeed different among the RANS models. However, it is difficult to verify which one is better because these variables are not direct outputs of the LES model and require additional modeling choices, see answer 1 to Reviewer 1.

2. In the comparison to LES (Section 4.1), it's not clear to this reader how the parameters in Table 2 have been estimated. My concern is that since the LES is being used for parameter calibration, that subsequently assessing accuracy of the models against the same data is a statistical "inverse crime", and not really informative with respect to model accuracy.

We think this is a misunderstanding. The inflow of wind farm simulations is always calibrated with a reference in order to apply the appropriate boundary conditions. This could be a met. mast or a high fidelity model. In some cases, only the hub height wind speed and turbulence intensity are known and set. In other cases, an entire reference profile is used. In the present article, we try to mimic the LES inflow profiles. The RANS models are subsequently tested against the LES results for the single wake

flow, which does not include additional calibration.

**Didactic improvements**

1. Convective ABLs are not considered here, please explain why not.
   Modeling convective ABLs with a steady-state model as RANS is not trivial, and is a subject which has defied solution for several decades (still an active research area). One could add an additional source as performed in a previous work van der Laan et al. (2020). This can improve the results the atmospheric surface layer but nonphysical very large ABLs can be obtained as well. We have added more information in the introduction.

2. The Brunt Vaisala frequency is introduced without comment. Explain what it is, and how it relates physically to the height of the ABL.
   We have added a clarification in Sect. 2.3.

3. A plot of the RANS-Theta profile vs the RANS-N profile (or its gradient) in the model discussion would be instructive.
   We added the implied temperature profile of the RANS-$N$ in Fig. 2 and added a discussion in Sect. 4.1.

4. Typos: l108 - lowercase theta.
   We now use an uppercase symbol everywhere.

**References**

E L Hodgson, M Souaiby, N Troldborg, F Porté-Agel, and S J Andersen. Cross-code verification of non-neutral abl and single wind turbine wake modelling in les. *Journal of Physics: Conference Series*, 2505 (1):012009, may 2023. doi: 10.1088/1742-6596/2505/1/012009. URL `https://dx.doi.org/10.1088/1742-6596/2505/1/012009`.

M. P. van der Laan and S. J. Andersen. The turbulence scales of a wind turbine wake: A revisit of extended k-epsilon models. *Journal of Physics: Conference Series*, 1037(072001):1–10, 2018.

M. P. van der Laan, M. Kelly, R. Floors, and A. Peña. Rossby number similarity of an atmospheric rans model using limited-length-scale turbulence closures extended to unstable stratification. *Wind Energy Science*, 5(1):355–374, 2020. doi: 10.5194/wes-5-355-2020.

M. P. van der Laan, O. García-Santiago, M. Kelly, A. Meyer Forsting, C. Dubreuil-Boisclair, K. Sponheim Seim, M. Imberger, A. Peña, N. N. Sørensen, and P.-E. Réthoré. A new rans-based wind farm parameterization and inflow model for wind farm cluster modeling. *Wind Energy Science*, 8(5):819–848, 2023. doi: 10.5194/wes-8-819-2023. URL `https://wes.copernicus.org/articles/8/819/2023/`.

Y. T. Wu and F. Porté-Agel. Large-Eddy Simulation of Wind-Turbine Wakes: Evaluation of Turbine Parametrisations. *Boundary-Layer Meteorology*, 138:345–366, 2011.